# Structures reveal a key mechanism of WAVE regulatory complex activation by Rac1 GTPase

Bojian Ding [1,10], Sheng Yang [2,8,10], Matthias Schaks [3,4,9], Yijun Liu[2], Abbigale J. Brown[2], Klemens Rottner [3,4,5], Saikat Chowdhury [1,6,7] ✉ & Baoyu Chen [2] ✉

The Rho-family GTPase Rac1 activates the WAVE regulatory complex (WRC) to drive Arp2/3 complex-mediated actin polymerization in many essential processes. Rac1 binds to WRC at two distinct sites—the A and D sites. Precisely how Rac1 binds and how the binding triggers WRC activation remain unknown. Here we report WRC structures by itself, and when bound to single or double Rac1 molecules, at ~3 Å resolutions by cryogenic-electron microscopy. The structures reveal that Rac1 binds to the two sites by distinct mechanisms, and binding to the A site, but not the D site, drives WRC activation. Activation involves a series of unique conformational changes leading to the release of sequestered WCA (WH2-central-acidic) polypeptide, which stimulates the Arp2/3 complex to polymerize actin. Together with biochemical and cellular analyses, the structures provide a novel mechanistic understanding of how the Rac1-WRC-Arp2/3-actin signaling axis is regulated in diverse biological processes and diseases.

The Wiskott-Aldrich Syndrome Protein (WASP) family proteins play a central role in promoting Arp2/3 complex-mediated actin assembly in a wide range of processes, including cell migration and intracellular vesicle trafficking[1–3]. These proteins share a conserved C-terminal WCA (WH2-central-acidic) sequence, which can bind to and stimulate the Arp2/3 complex to produce branched actin networks at membranes[1–3]. Most WASP-family proteins are inhibited in the basal state[1]. Inhibition is achieved by keeping their WCA sequence sequestered either *in cis* within a single polypeptide chain, as in WASP and N-WASP, or in trans within large multi-protein complexes, as in WAVE and WASH[3–11]. A variety of upstream signals, including ligand binding (e.g., GTPases, inositol phospholipids, membrane receptors, and scaffolding proteins) and post-translational modifications (e.g., phosphorylation and ubiquitination), often act cooperatively in the cell to relieve the inhibition and simultaneously recruit WASP-family proteins to their target membrane locations to promote actin polymerization[3,9,11–18].

The WASP-family member WAVE exists exclusively in a 400-kDa, hetero-pentameric assembly named the WAVE Regulatory Complex (WRC)[3,19]. Essential to most eukaryotic organisms, the WRC plays a key role in promoting actin polymerization at plasma membranes and producing sheet-like membrane protrusions known as lamellipodia, commonly found at the leading edge of migrating cells[2,5]. Genetic mutations in various subunits of the WRC are frequently associated

[1]Department of Biochemistry and Cell Biology, Stony Brook University, 100 Nicolls Road, Stony Brook, NY 11794, USA. [2]Roy J. Carver Department of Biochemistry, Biophysics & Molecular Biology, Iowa State University, 2437 Pammel Drive, Ames, IA 50011, USA. [3]Division of Molecular Cell Biology, Zoological Institute, Technische Universität Braunschweig, Spielmannstrasse 7, 38106 Braunschweig, Germany. [4]Department of Cell Biology, Helmholtz Centre for Infection Research, Inhoffenstrasse 7, 38124 Braunschweig, Germany. [5]Braunschweig Integrated Centre of Systems Biology (BRICS), Rebenring 56, 38106 Braunschweig, Germany. [6]CSIR-Centre for Cellular and Molecular Biology, Hyderabad, Telangana 500007, India. [7]Academy of Scientific and Innovative Research (AcSIR), Ghaziabad, Uttar Pradesh 201002, India. [8]Present address: Target & Protein Sciences, Janssen R&D, Johnson & Johnson, 1400 McKean Rd, Spring house, PA 19477, USA. [9]Present address: Soilytix GmbH, Dammtorwall 7 A, 20354 Hamburg, Germany. [10]These authors contributed equally: Bojian Ding, Sheng Yang. ✉e-mail: saikat@csirccmb.org; stone@iastate.edu

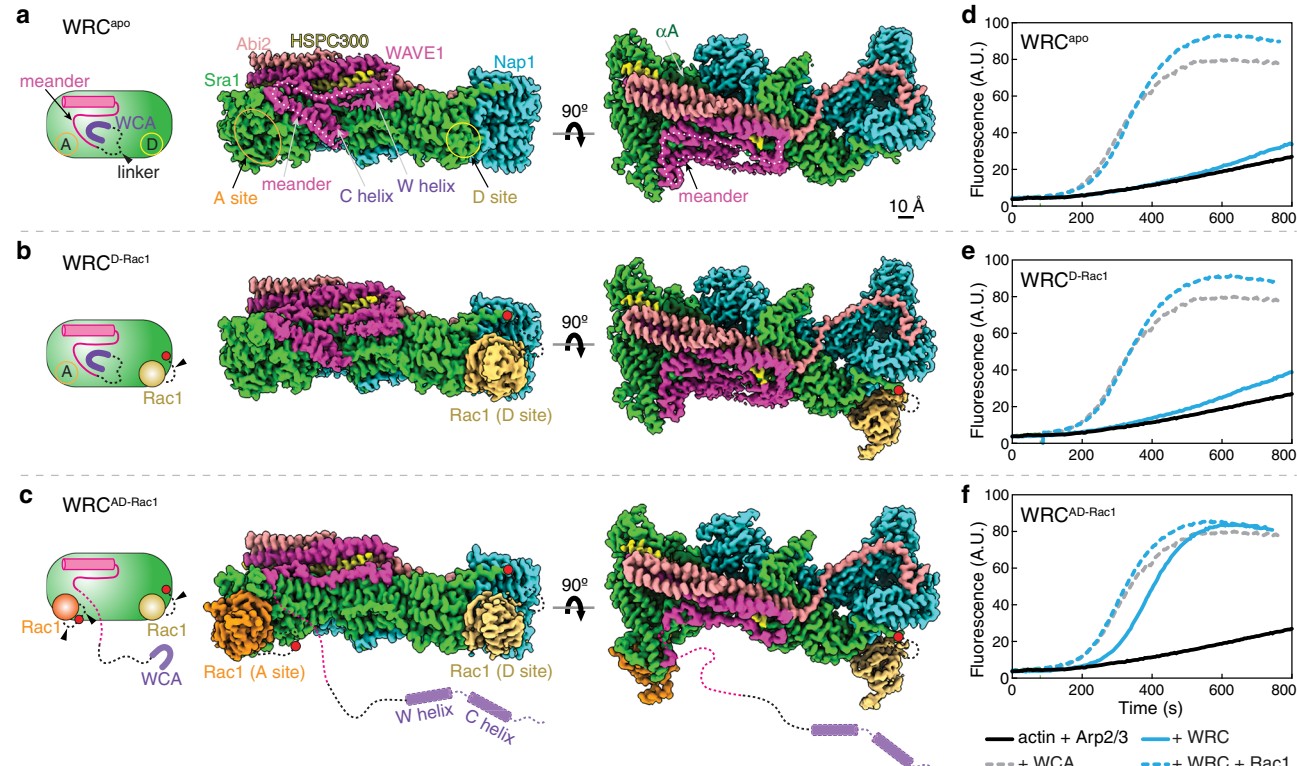

**Fig. 1 | Cryo-EM structures of the WRC in different Rac1-bound states.**
**a–c** Schematic and cryo-EM density of the indicated WRCs. Black dotted lines indicated by arrowheads are flexible peptide linkers tethering Rac1 to the WRC. The meander sequence is traced by white dotted lines. Other dotted lines and cylinders in the WRC[AD-Rac1] structure refer to sequences of which the densities are not observed in WRC[AD-Rac1], but present in WRC[apo] and WRC[D-Rac1]. Red dots indicate locations to which Rac1 is tethered. **d–f** Pyrene-actin polymerization assays measuring activities of the indicated WRCs used in cryo-EM. Reactions use the NMEH20GD buffer (see Methods) and contain 3.5 μM actin (5% pyrene-labeled), 10 nM Arp2/3 complex, 100 nM WRC230WCA or WAVE1 WCA, and/or 6 μM Rac1[QP]. A. U. for arbitrary units. Results are representative of at least two independent repeats. Source data for **d–f** are provided as a Source Data file.

with human diseases, including neurodevelopmental disorders, immune syndromes, and many types of cancer[3,19].

WRC by itself exists in an autoinhibited state[7–9]. Previous crystal structures of a minimal, inhibited WRC (WRC[xtal]) revealed the overall structural organization and the inhibition mechanism[9,13]. The studied complex is composed of a trimer formed by WAVE1, Abi2, and HSPC300, and a dimer formed by Sra1 and Nap1 subunits (Fig. 1a). WAVE1 has a conserved sequence of ~100 amino acids (a.a.) defined as the "meander" sequence, which meanders across the surface of Sra1 as a loose collection of loops and short helices[9] (Fig. 1a). The meander region and a conserved surface on Sra1 collectively comprise the WCA-binding site, which sequesters the WCA from accessing the Arp2/3 complex[9].

A large variety of ligands can recruit WRC to membranes through direct interactions and/or simultaneously activate it[19]. Among these ligands, the Rho-family GTPase Rac1 is the ubiquitous activator of WRC[19]. Prior studies have shown that direct binding of Rac1 to the WRC is necessary and, in many cases, sufficient to drive WRC activation[6,12–14,20,21]. Determining how Rac1 binds to and activates WRC is key to understanding the regulation and function of WRC-mediated signaling in diverse cellular processes.

Previous biochemical and low-resolution cryo-EM studies proposed two distinct Rac1-binding sites on WRC[9,14]. Both sites were mapped on conserved surfaces of Sra1, which are separated by ~100 Å on the opposite ends of the WRC. The site adjacent to the WCA was referred to as the A site, and the one distant from the WCA as the D site[14] (Fig. 1a). The two sites thus allow simultaneous binding of two individual Rac1 molecules, albeit with distinct affinities, with the D site binding ~40–100 times stronger than the A site[14]. Mutating key residues at either site abolished WRC activation by Rac1 in pyrene-actin

polymerization assays, suggesting both sites are critical for WRC activation[14]. In vivo, however, the A site seemed to play a more important role in promoting lamellipodia formation in both mammalian and amoeba cells, despite its much lower affinity for Rac1[21]. Mutating the A site almost completely abolished lamellipodia formation, while mutating the D site did not eliminate lamellipodia formation, but compromised their general morphology[21]. Without high-resolution structures of the WRC bound to Rac1, it remains unclear how Rac1 binds the two sites and, more importantly, how Rac1 binding triggers WRC activation.

Here we report WRC structures determined to ~3 Å resolutions by single-particle cryogenic-electron microscopy (cryo-EM) in three different states: without Rac1, with Rac1 only bound to the D site, and with Rac1 molecules bound to both A and D sites simultaneously. The structures reveal in detail how Rac1 interacts with both sites, and how Rac1-WRC interaction drives WRC activation. We find Rac1 uses two distinct mechanisms to bind the two sites, and only binding to the A site, but not to the D site, directly contributes to WRC activation. Rac1 binding to the A site flattens the interaction surface and allosterically destabilizes a conserved region in the meander sequence of WAVE1 critical for autoinhibition. These conformational changes release the sequestered WCA sequence, making it accessible to Arp2/3 complex. Coupled with biochemical and cellular studies, the new structures resolve a central mechanism of WRC activation, a key step in Arp2/3 complex-mediated actin filament assembly in the cell.

## Results
### Structural solutions to understand Rac1 binding to WRC
Prior structural studies of WRC led to the identification of the A and D sites, but could not determine the binding or activation mechanism

due to several limitations[9,14]. The crystal structures used a "minimal WRC" (WRC$^{xtal}$) that lacked the polypeptide region (a.a. 187-230 was deleted from WAVE1) necessary for activation, and hence could not be activated by Rac1[9,14]. The cryo-EM structure of WRC bound to a single Rac1 lacked WCA (WRC$^{230\Delta WCA\text{-}Rac1}$), and its limited resolution could not precisely delineate how Rac1 binds to the D site, or whether D site Rac1 binding causes activation[14]. Moreover, many questions about the A site remained open due to the lack of structural information, including exactly where the proposed A site is, how Rac1 binds to it, and whether or how Rac1 binding to the A site activates WRC.

A major challenge in addressing the aforementioned questions is the weak affinity between Rac1 and WRC, especially at the A site[14], which prevents the formation of stable Rac1-WRC complexes amenable for structural studies. To overcome this challenge, we developed new strategies here to stabilize Rac1 binding. To optimize Rac1 binding to the D site, we used a different tethering strategy than what was used in WRC$^{230\Delta WCA\text{-}Rac1}$ in the previous study[14] (Supplementary Fig. 2a, b). Firstly, we tethered Rac1 to the C-terminus of Sra1. This avoids tethering Rac1 to WAVE1 as in the previous strategy, which would directly perturb WCA at the C-terminus of WAVE1. Secondly, due to the close proximity of Sra1 C-terminus to the D site, we used a significantly shorter, flexible peptide linker (shorter by 71 a.a.) to tether Rac1 to WRC, which keeps Rac1 closer to the D site (herein referred to as WRC$^{D\text{-}Rac1}$; Fig. 1b, red dot). In both the previous[14] and new constructs, we use a Rac1 harboring Q61L/P29S mutations (hereafter referred to as Rac1 or Rac1$^{QP}$ interchangeably), in which Q61L stabilizes GTP binding to Rac1, and P29S (an oncogenic mutation identified in melanoma patients[22,23]) enhances Rac1 binding to WRC[14].

Stabilizing Rac1 binding to the A site was more challenging due to its lower affinity. After exploring several different strategies, one of the approaches favored stable binding of Rac1 to the A site (Fig. 1c and Supplementary Fig. 3a). In this strategy, we inserted a Rac1$^{QP}$ to the middle of a non-conserved surface loop of Sra1, between Y423/S424 of a.a. 418-432, using two separate flexible linkers. Additionally, we kept a Rac1 tethered to the C-terminus of Sra1 as in WRC$^{D\text{-}Rac1}$. However, instead of using Rac1$^{QP}$, we used Rac1$^{P29S}$ for the D site as we found Rac1$^{P29S}$ could be readily loaded with GDP or GTP (or GMPPPNP, a nonhydrolyzable GTP analog), while Rac1$^{QP}$ remained locked to GTP in the same condition (Supplementary Fig. 3e). This unique feature allowed us to find that (1) Rac1 tethered to the A site was sufficient to activate the WRC (when Rac1$^{P29S}$ at the D site was loaded with GDP); and (2) the D-site bound Rac1 could further improve WRC activation by the A site bound Rac1 (when Rac1$^{P29S}$ at the D site was loaded with GMPPNP) (Supplementary Fig. 3d). We refer to this construct as WRC$^{AD\text{-}Rac1}$. In both WRC$^{D\text{-}Rac1}$ and WRC$^{AD\text{-}Rac1}$, we did not remove WCA to enhance Rac1 binding, which is contrary to what was previously done for the WRC$^{230\Delta WCA\text{-}Rac1}$ construct[14]. Instead, we kept WCA intact to unbiasedly validate the active state of the WRC-Rac1 complexes, both by activity measurements and by the existence of WCA density in reconstructed maps (Fig. 1).

Using cryo-EM we determined structures of both WRC$^{D\text{-}Rac1}$ and WRC$^{AD\text{-}Rac1}$, as well as WRC$^{apo}$ (WRC without tethered Rac1) to ~3 Å resolutions (Fig. 1, Supplementary Fig. 1-3, and Supplementary Table 3). The overall structures of WRC in these three complexes were similar to the previous crystal structure (WRC$^{xtal}$)[9], with a root-mean-square deviation (r.m.s.d.) of 0.83-1.22 Å (Supplementary Fig. 1a). Except for local conformational changes caused by A site Rac1 binding as described in detail below, Rac1 binding to either A or D site did not cause large-scale, global conformational changes to WRC. The additional sequence in WAVE1, a.a. 187-230, which was not included in the previous WRC$^{xtal}$ construct, did not show any density in the cryo-EM reconstructions, suggesting that this region is disordered in WRC. Additionally, the structures unambiguously determined that the N-terminal helix of Sra1 (αA, a.a. 5-22) belonged to the same complex, instead of a neighboring WRC as suggested by the previous structural

study using X-ray crystallography[9] (Fig. 1a, dark green, and Supplementary Fig. 1b, c).

Comparison between the WRC$^{apo}$, WRC$^{D\text{-}Rac1}$, and WRC$^{AD\text{-}Rac1}$ structures clearly reveal that Rac1 binding to the A site, but not the D site, leads to WRC activation. In WRC$^{D\text{-}Rac1}$, D site Rac1 binding did not cause obvious conformational changes; neither did it destabilize the meander sequence or the W and C helices (Fig. 1b vs. 1a and Supplementary Fig. 2). By contrast, in WRC$^{AD\text{-}Rac1}$, the density for the W and C helices and part of the meander region were not observed in the reconstructed map, suggesting that upon Rac1 binding to the A site, these sequences were destabilized and released from the WRC (Fig. 1c). Consistent with these observations, the pyrene-actin polymerization assay showed both WRC$^{apo}$ and WRC$^{D\text{-}Rac1}$ were autoinhibited and could be activated by the addition of free Rac1 (Fig. 1d, e, blue curves), whereas WRC$^{AD\text{-}Rac1}$ was basally active (Fig. 1f, solid blue curve). Furthermore, a WRC with Rac1$^{P29S}$ only tethered to the A site, but not the D site (herein referred to as WRC$^{A\text{-}Rac1}$), was also basally active in a nucleotide-dependent manner (Supplementary Fig. 3d). Together, these results confirm that A site binding, but not D site binding, causes WRC activation.

It is important to note that densities corresponding to the flexible peptide linkers used to tether Rac1 to WRC were not visible in the reconstructed maps, not did we observe perturbations of local structures surrounding the tethering points (Supplementary Fig. 2c and 3c). These observations indicate the linkers only facilitated binding by increasing the local concentration of Rac1, but did not influence or perturb the native structures or caused artefactual conformational states. Additionally, the poise of Rac1 at both A and D sites is consistent with how GTPases interact with effectors in general and is compatible with WRC orientation at plasma membranes[14] (Supplementary Fig. 4). Furthermore, the WRCs used in our structural and biochemical studies behaved similarly during chromatographic purifications, showing no sign of mis-assembly or aggregation (Supplementary Tables 1, 2; Supplementary Fig. 5). Together with complementary structural, biochemical, and cellular analysis described below, our structures represent the native binding states of Rac1 to WRC.

## Interactions between Rac1 and the D site of WRC

Both WRC$^{D\text{-}Rac1}$ and WRC$^{AD\text{-}Rac1}$ show nearly identical structures of Rac1 binding to the D site (0.53 Å r.m.s.d.). Compared to WRC$^{apo}$, WRC$^{D\text{-}Rac1}$ does not show major structural differences (0.46 Å r.m.s.d. over the whole complexes; Fig. 1a, b). Importantly, the density for structural elements crucial to WRC activation, including the meander sequence and WCA, remains intact and virtually identical to WRC$^{apo}$, suggesting that D site Rac1 binding does not directly promote WRC activation (Fig. 1b). This is consistent with results from pyrene-actin polymerization assays, which show WRC$^{D\text{-}Rac1}$ is autoinhibited (Fig. 1e, solid blue curve). That this inhibition can be relieved by free Rac1 emphasizes that WRC$^{D\text{-}Rac1}$ is a functional complex (Fig. 1e, dashed blue curve),

The D site engages with Rac1 mainly through a largely hydrophobic surface of ~550 Å$^2$ presented by a.a. 961-978 of Sra1, which contains a helix-turn-loop structure formed by the H11b1 helix and the loop between H11b1 and H11a helices (Fig. 2a, b and Supplementary Fig. 3b). Solvent-exposed residues, including R961, R964, Y967, E974, F975, and H978, form multiple hydrophobic, π-π stacking, and polar interactions with several parts of Rac1, including a small portion (a.a. 36-39) of Switch I motif, the helical part (a.a. 64-66) of Switch II motif, and W56 at the N-terminus of Switch II (Fig. 2b and Supplementary Fig. 6a-d,p).

The peptide backbone of contacting residues in either Sra1 or Rac1 does not show major structural changes upon Rac1 binding, except a shift of ~1-1.5 Å of the helix (a.a. 64-70) in Switch II of Rac1 towards Sra1 (Fig. 2d, black arrow). Several side chains, however, undergo significant rotations to establish the binding. Among them, the side chains of Y967 on Sra1 and F37 on Rac1 undertake a dramatic

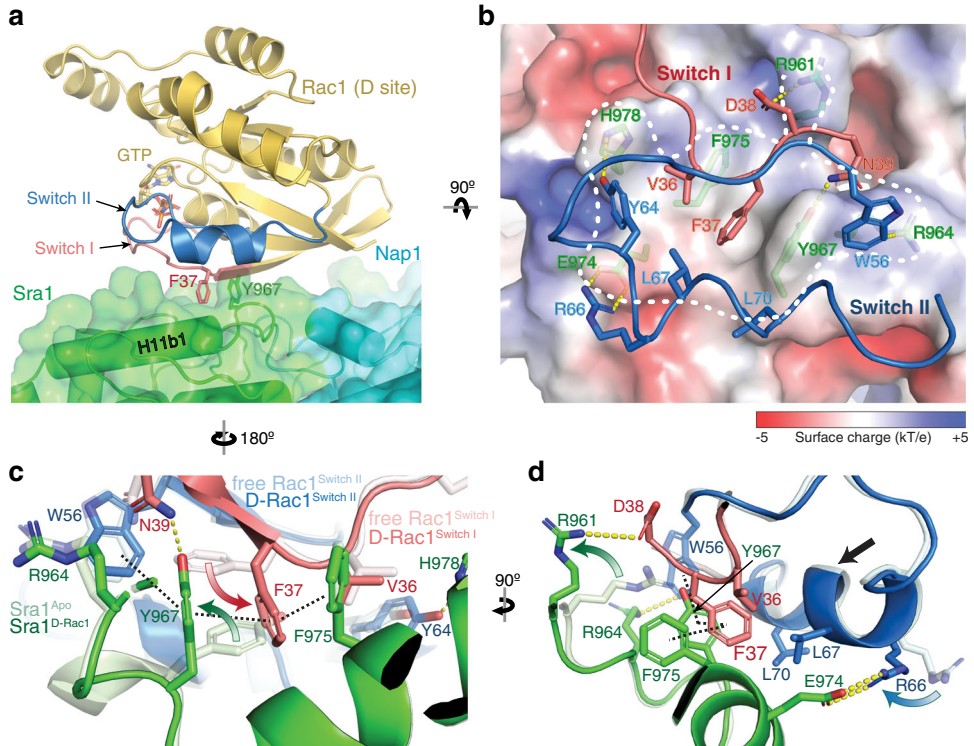

**Fig. 2 | Interactions mediating Rac1 binding to the D site. a** Side view of the overall structure of Rac1 (cartoon, gold) binding to the D site (semitransparent surface, green). F37 and Y967 side chains are shown as reference points. **b** Top view and semitransparent surface charge representation of the D site (calculated using APBS in Pymol[45]), showing key interactions between Sra1 and Rac1. Yellow dotted lines indicate polar interactions. White dashed line indicates binding surface boundary. For clarity, the backbones of Switch I and II are shown as loops. **c, d** Structural comparison of Rac1 and the D site in the bound (dark colors) and unbound (light colors, PDB 3SBD for Rac1) states. Curved arrows indicate side chain flipping upon Rac1 binding. Straight arrow indicates translation of polypeptide backbone. Black dashed lines indicate the interdigitated π-π stacking.

rotation of -80° and 110°, respectively, creating a tight π-π stacking between the two residues (Fig. 2c, curved arrows, and Supplementary Fig. 6a). Further, but less ideal stacking of two additional aromatic residues, Rac1$^{W56}$ and Sra1$^{F975}$, which flank the Sra1$^{Y967}$-Rac1$^{F37}$ core, creates an "interlock" to stabilize Rac1 binding (Fig. 2c, d, black dotted lines).

In addition to the π-π stacking, the conformation of Rac1$^{F37}$ is stabilized by a hydrophobic pocket formed by Rac1$^{V36,L67,L70}$ and Sra1$^{P963,I972,Y967,G971,F975}$ (Supplementary Fig. 6d). This Sra1$^{Y967}$-Rac1$^{F37}$ core interaction is further stabilized by several polar interactions at the periphery, including a cation-π interaction between Sra1$^{R964}$ and Rac1$^{W56}$, hydrogen bonding between Sra1$^{Y967}$ and Rac1$^{N39}$ and between Sra1$^{H978}$ and Rac1$^{Y64}$, and two salt bridges, one between Sra1$^{E974}$ and Rac1$^{R66}$ and the other between Sra1$^{R961}$ and Rac1$^{D38}$ (Fig. 2b–d, and Supplementary Fig. 6a–c,p). In particular, the guanidino group of Rac1$^{R66}$ swings forward by ~4 Å to engage with Sra1$^{E974}$, while the side chain of Sra1$^{R961}$ swings away to avoid steric clashes with Rac1 and engage with Rac1$^{D38}$ (Fig. 2d, curved arrows, Supplementary Fig. 6b, c). These structures are consistent with observations made in prior studies[14,21], where substituting the key residues at the D site, including Y967A, G971W, R961D/P963A/R964D, and E974A/F975A/H978A/Q979A, all disrupted Rac1 binding to WRC.

### General overview of Rac1 binding to the A site of WRC

When the A site is occupied by Rac1, we noticed that the A site, the meander region, and the WCA helices have undergone significant local conformational changes. This is in clear contrast to what is observed when Rac1 binds to the D site. Particularly, no densities are observed for WCA helices in WRC$^{AD-Rac1}$, which indicates Rac1 binding to the A site activates the WRC (Fig. 1c). This is consistent with the biochemical observations that WRCs with Rac1 bound to the A site are always active,

irrespective of Rac1 binding to the D site (Fig. 1f and Supplementary Fig. 3d). In the following two sections, we will answer two important questions: (1) how Rac1 binds to the A site; (2) how Rac1 binding leads to WRC activation.

### Interactions between Rac1 and the A site of WRC

The A site constitutes an extensive surface on the N-terminal region of Sra1 (also called the DUF1394 domain; Supplementary Fig. 3b). Distinct from the D site, which is flat, relatively small, and largely hydrophobic, the A site is concave, nearly two times larger (-1138 Å²) than the D site, and mostly positively charged (Fig. 3a, b). On the Sra1 side, the binding involves a.a. 91–108 (helix H1b1 and loop L2 that connects H1a to H1b1) and a.a. 176–215 (mainly αB and αC helices) (Fig. 3a). On the Rac1 side, the interactions involve the end of the α1 helix (a.a. 23–25), most of Switch I (a.a. 26–37), the β2-β3 beta-sheet connecting Switch I and Switch II (a.a. 38–55), and the beginning of Switch II (a.a. 56–70) (Fig. 3a, b). Majority of the interactions differ from those between Rac1 and its inhibitor CYRI-B (CYFIP-related Rac1 interactor, or FAM49B), which shares a remote homology with the A site region of Sra1[24–26] (Supplementary Fig. 6p–s).

Given the low affinity, it is surprising that A site Rac1 binding is mediated by numerous, mostly polar interactions, including 17 hydrogen bonds, 3 salt bridges, and several hydrophobic interactions (Fig. 3b and see Supplementary Fig. 6e–m,p for more details). Among them, the interactions clustered around Sra1$^{R190}$ seem to play a particularly important role. Sra1$^{R190}$, which is strictly conserved in almost all organisms[9], is anchored within an acidic pocket on the Rac1 surface formed partially by the Switch I loop (Fig. 3c). The anchoring of R190 is stabilized by the formation of a salt bridge with Rac1$^{E31}$, which is further bolstered by many interactions at the ridges of the pocket, including salt bridges between Sra1$^{K189}$ and Rac1$^{D38}$, and Sra1$^{R104}$ and Rac1$^{E31}$,

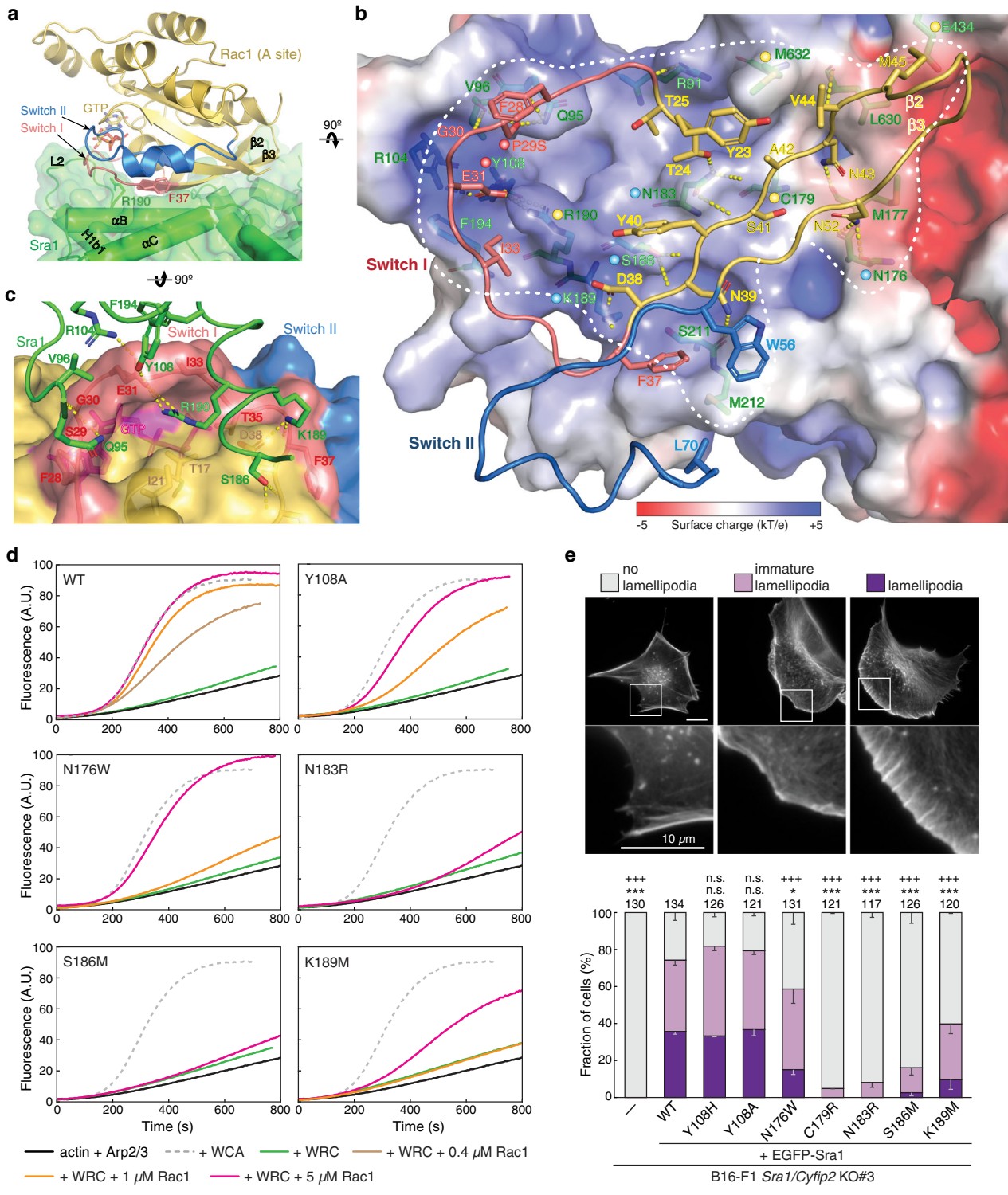

hydrogen bonding between Sra1$^{K189}$ and Rac1$^{F37\ CO}$, Sra1$^{Y108}$ and Rac1$^{E31}$, Sra1$^{V96\ CO}$ and Rac1$^{G30\ NH}$, and Sra1$^{Q95,\ V96\ CO}$ and Rac1$^{S29}$ (details of P29S mutation in Rac1 is described in later section), as well as a hydrophobic interaction between Sra1$^{F194}$ and Rac1$^{I33}$ (Fig. 3c and Supplementary Fig. 6e–g,p). This Arginine anchor is unique to Sra1 (Fig. 3c) and CYRI-B (Supplementary Fig. 6s), as it is not observed in other reported Rac1 binding proteins. Consistent with the structural analysis, mutating R190 to Aspartic acid (R190D) in previous studies disrupted Rac1 binding in vitro and abolished WRC activity in lamellipodia formation[9,21].

In addition to R190D, the structure also explains another previously studied mutation at the A site, C179R, which is located distant from R190, but similarly abolished WRC activity[9,14,21]. Limited to small side chains in all examined organisms[9], C179 is tightly packed against a concave pocket on Rac1 (Supplementary Fig. 6n). Although it does not form any specific interactions with Rac1, mutating C179 to the long-chain residue Arginine would cause steric clashes to disrupt Rac1 binding to the A site.

To further understand the contributions of individual interactions, we mutated several additional conserved contacting residues at

**Fig. 3 | Interactions mediating Rac1 binding to the A site. a** Side view of the overall structure of Rac1 binding to the A site, using the same color scheme as in Fig. 2. F37 and R190 side chains are shown as reference points. **b** Top view and semitransparent surface charge representation of the A site, showing key interactions between Sra1 and Rac1. Yellow dotted lines indicate polar interactions. White dashed line indicates binding site boundary. For clarity, the backbone of Rac1 Switch I−β2−β3−Switch II sequence mediating the binding is shown as loops. Dots of different colors indicate residues of which mutations were involved in human disease (red), previously designed and shown to disrupt Rac1 binding (yellow), or newly introduced in this work (blue). **c** Semitransparent surface representation of the Rac1 surface, showing how Sra1[R190] fits into a deep pocket in Rac1 and how it is supported by interactions surrounding the rim of the pocket. **d** Pyrene-actin polymerization assays measuring the activities of WRCs carrying indicated mutations at the A site. Reactions use the NMEH20GD buffer (see Methods) and contain 3.5 μM actin (5% pyrene-labeled), 10 nM Arp2/3 complex, 100 nM WRC230WCA or

WAVE1 WCA, and/or indicated amounts of Rac1[QP]. **e** Representative fluorescence images and quantification of lamellipodia formation in B16-F1 *Sra1/Cyfip2* double KO#3 cells transfected with indicated EGFP-Sra1 variants and stained by phalloidin for F-actin. Statistical significance was assessed from three repeats for differences between cells transfected with WT (wild type) *vs.* no (-) or indicated mutant constructs concerning cell percentages displaying "no lamellipodia" phenotype (*$p < 0.05$; ***$p < 0.001$) and with "lamellipodia" phenotype (+++$p < 0.001$) using one-way ANOVA with Dunnett's post hoc test correcting for multiple comparisons. n.s.: not statistically significant. Error bars represent standard errors of means. Cell numbers used for the quantification are shown on top of each column. Exact $p$-values (WT vs. X) are No lamellipodia: no (-): <0.0001, Y108H: 0.4733, Y108A: 0.8198, N176W: 0.0240, C179R < 0.0001, N183R < 0.0001, S186M < 0.0001, K189M < 0.0001; and Lamellipodia: no (-): <0.0001, Y108H: 0.9706, Y108A: 0.9995, N176W < 0.0001, C179R < 0.0001, N183R < 0.0001, S186M < 0.0001, K189M < 0.0001. Source data for **d**, **e** are provided as a Source Data file.

the A site, including N176W, N183R, S186M, and K189M. They form hydrogen bonds or salt bridges with N52, T24/S41 NH/CO, N39 NH/CO, and D38/F37 CO in Rac1, respectively (Fig. 3b and Supplementary Fig. 6g, h, k). All mutations disrupted WRC activation in vitro (Fig. 3d) and reduced lamellipodia formation upon re-introduction of corresponding variants into B16-F1 *Sra1/Cyfip2* double knock-out (KO) cells[21] (Fig. 3e and Supplementary Fig. 7b). The mutations did not affect WRC purification in vitro and rescued WRC expression in the cell (Supplementary Fig. 5e-j and 7c), suggesting they did not interfere with protein folding or complex assembly. It is intriguing that the effects of these mutants varied based on their relative locations in the A site. N183R and S186M, which are located at the center of the A site, strongly impaired WRC function, while N176W and K189M, which are located at the periphery of the A site, had milder effects (Fig. 3b, d, e). This suggests individual interactions have different contributions to the overall binding to Rac1, with residues at the center of the A site having major contributions. It is worth noting that the efficiency of these mutants in disrupting WRC's function in vitro correlates well with their extent in compromising lamellipodia formation in cells (Fig. 3d vs. e), thus supporting that the biochemical activity of Rac1 in binding and activating the WRC is directly correlated with WRC activity in promoting lamellipodia formation.

In addition to the aforementioned mutations, the WRC[AD-Rac1] structure explains the phenotype of several mutations found in human patients. P29S in Rac1 was initially identified as one of the major somatic mutations in human melanoma, shown to enhance Rac1 binding to various effector proteins, including PAK1 (p21 protein activated kinase 1), MLK3 (mixed-lineage kinase 3), and the WRC[14,22,23]. For this reason, we included the P29S mutation in our constructs in order to strengthen Rac1 binding[14]. Our structure shows that P29S provides additional hydrogen bonding with Sra1[Q95] and Sra1[V96 CO], which explains why this melanoma-causing mutation promoted WRC binding and activation[14] (Fig. 3b and Supplementary Fig. 6e).

Furthermore, Y108H in the Sra1 homolog Cyfip2 (which shares 88% sequence identity with Sra1 and is analogously incorporated into a WRC) is one of the hotspot mutations found in patients with developmental and epileptic encephalopathy-65 (DEE-65)[27]. Unlike other hotspot mutations in DEE-65 (described later), Y108H does not seem to directly affect the meander or WCA sequence. Our structure shows that Sra1[Y108] forms hydrogen bonds with Rac1[E31] (one of the aforementioned interactions at the ridges of the acidic pocket stabilizing R190 anchoring) (Fig. 3b, c and Supplementary Fig. 6f). Replacing this Tyr to a His, which has a similar size as Tyr, but is more polar and positively charged, may further enhance this polar interaction to promote Rac1 binding and WRC activation. Consistent with this prediction, Y108H in Sra1 mildly sensitized WRC activation by Rac1 (Supplementary Fig. 7a), whereas Y108A slightly reduced the sensitivity (Fig. 3d and Supplementary Fig. 7a). The effect of Y108H or Y108A was

subtle, suggesting the Sra1[Y108]-Rac1[E31] interaction has a limited contribution to Rac1 binding. In fact, in our complementation assays using B16-F1 *Sra1/Cyfip2* KO cells, where Rac1 expression was not disrupted[21], neither Y108H nor Y108A significantly affected WRC-mediated lamellipodia formation (Fig. 3e and Supplementary Fig. 7b,c). In contrast, in previously published results using a sensitized cell line, *Sra1/Cyfip2/Rac1/2/3* KO B16-F1 cells (clone #3/4), where Rac1 expression was substantially reduced (but not completely eliminated) making lamellipodia formation more sensitive to Rac1-WRC interaction, Y108H mutation indeed enhanced WRC-dependent actin remodeling[28]. These observations provide an example that even a moderate enhancement of WRC activity in the cell by Y108H mutation can disrupt the finely tuned regulation of Arp2/3 complex-mediated actin assembly, ultimately manifesting as a neurological disease.

Having understood the underlying mechanism of Rac1 binding to the A site, we next examine how this binding leads to WRC activation.

## Rac1 binding to the A site causes activating conformational changes in WRC

Despite the extensive interactions between Rac1 and A site, why does Rac1 bind to the A site with low affinity? By comparing the WRC[D-Rac1] with WRC[AD-Rac1] structure, we find that the conformation of the A site in WRC[D-Rac1] is not compatible with Rac1 binding. In particular, Rac1 binding would directly clash into part of the L2 loop (Fig. 4a and Supplementary Fig. 8a). To accommodate Rac1 binding, the A site must undergo a major conformational change, which involves flattening the concave binding surface by ~8°. This flattening is caused by an outward rotation of several key structural elements at the A site, including H1b1 helix, N-terminus of H1b2, loop L2, and αB-loop-αC, relative to a pivot axis running roughly through R87 in the L2 loop, K178 in the loop between H2a and αB, and N124 in H1b2, while keeping the neighboring structures beyond the pivot axis, including H1a, most part of H1b2, H2a, H2b, and H8a, unchanged (Fig. 4a, b, black arrows, and Supplementary Fig. 8a−c). The cost of the A site undergoing this conformational change could antagonize Rac1 binding and reduce Rac1's affinity to the A site significantly.

Another major conformational change occurring simultaneously when Rac1 binds to the A site is the release of the C-terminal half of the meander sequence in WAVE1 (after Q130, except part of the α4 which is accounted for by a poorly defined density in WRC[AD-Rac1]), together with the W and C helices (Fig. 4b, dark magenta vs. light pink). The release of W and C helices explains why WRC[AD-Rac1] is basally activated (Fig. 1c,f). In addition to the release of the C-terminal half of the meander sequence and WCA, the α2 helix of WAVE1 in the N-terminal half of the meander sequence partially unfolds and collapses towards where the W helix is originally located (Fig. 4b). Since α2 interacts with the W helix, which contributes to the sequestering of WCA, we believe the unfolding of α2 is a result of WCA release, leading to loss of support for the α2 conformation.

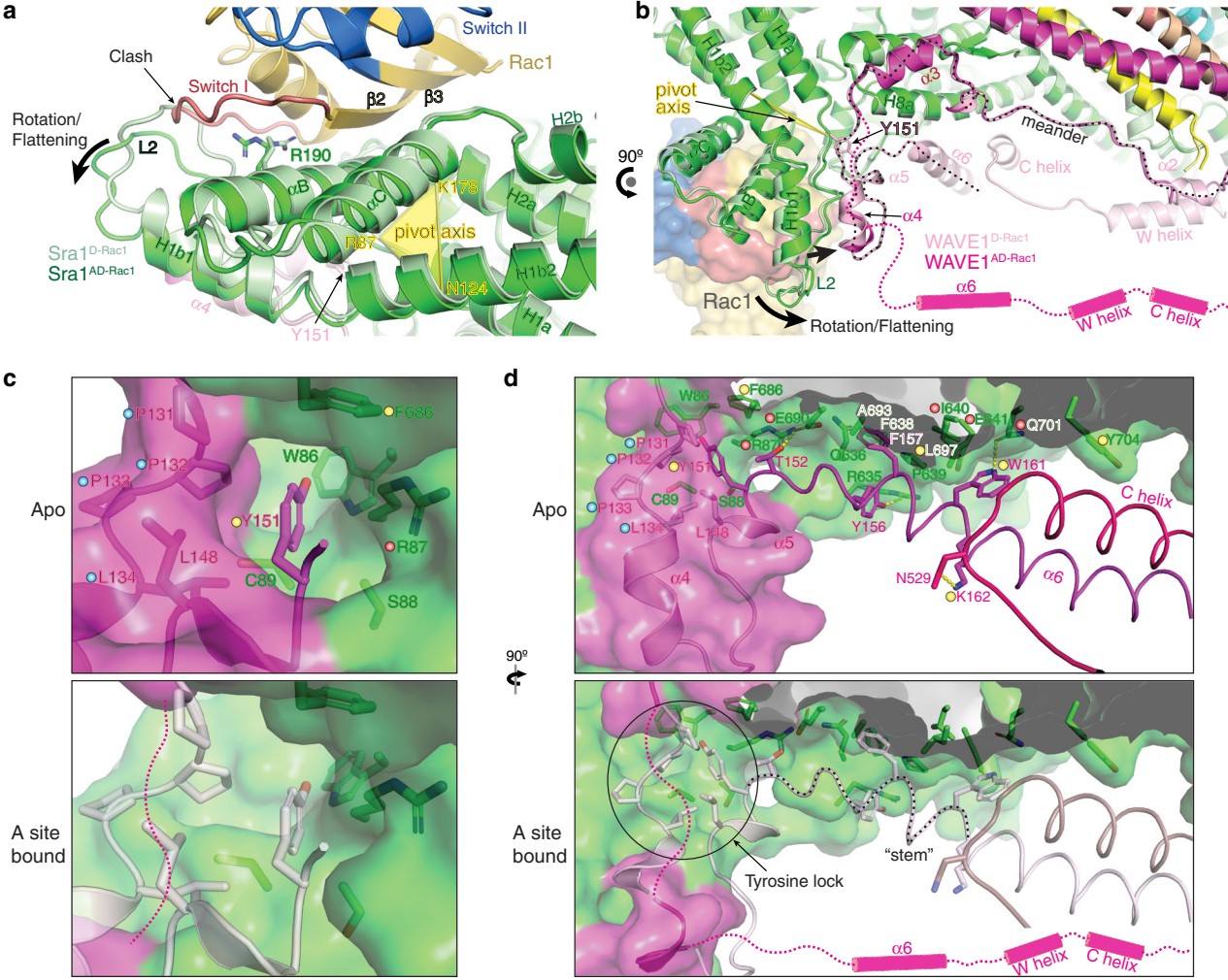

**Fig. 4 | Rac1 binding to the A site drives a conformational change to release WCA. a, b** Overlay of WRC^AD-Rac1^ (dark colors) and WRC^D-Rac1^ (light colors) structures showing conformational changes of the A site upon Rac1 binding. Sequences of which the densities are not observed in WRC^AD-Rac1^ structure, but are present in WRC^D-Rac1^ are indicated by magenta dashed lines and cylinders. The meander region in WAVE1 is traced by the black dotted line. **c, d** Comparison of the tyrosine lock and "stem" (traced by the black dotted line) region before and after Rac1 binding to the A site. Structures in light colors are from the unbound state in WRC^D-Rac1^ and used as reference point for the A-site bound state. Residues critical for stabilizing the tyrosine lock and "stem" components are labeled and shown in sticks. Dots of different colors indicate residues of which mutations were involved in human disease (red), previously designed and shown to disrupt WRC inhibition (yellow), or newly introduced in this work (blue).

Without a direct interaction, how does Rac1 binding to the A site allosterically release the meander sequence or WCA? Among the A-site elements that undergo rotations, H1b1 and L2 (a.a. 87–114) directly associate with the α4-loop-α5 component (a.a. 132–152) of the meander sequence in WAVE1 through a largely hydrophobic surface of ~686 Å² (Fig. 4b and Supplementary Fig. 8c–e). The rotation of H1b1 and L2 would simultaneously push α4-loop-α5 to rotate around the pivot axis, which aligns to a critical region of the meander sequence near residues P131 and Y151 (Fig. 4b and Supplementary Fig. 8b, c). At this region, Y151 inserts into a deep hydrophobic pocket formed by highly conserved residues from both Sra1 and WAVE1, analogous to a key being inserted into a lock (Fig. 4c, top). Based on this analogy, we herein refer to this region as the "tyrosine lock". Half of the tyrosine lock is contributed by Sra1, including W86/R87/S88/C89 from the L2 loop and F686 from the H8a helix, while the other half comes from WAVE1, including P131/P132/P133/L134 preceding α4 and L148 in α5 (Fig. 4c). In particular, P131/P132/P133 (or PPP) forms a rigid stereotypical left-handed polyproline II helix (PPII helix) that lines up the binding pocket (Fig. 4c). Rotating one side of the tyrosine lock (including S88, C89 in Sra1 and the rigid PPII helix in WAVE1), while keeping the other side stationary (including W67, R87, and F686 in

Sra1), would pinch the tyrosine lock and destabilize the binding of Y151 (Supplementary Movie 1).

The sequence immediately following Y151 (a.a. 151–161, herein referred to as the "stem") forms a series of highly conserved interactions with Sra1 and the C helix[9] (Fig. 4d), which are critical for keeping the meander sequence and WCA sequestered. Many missense mutations, either identified in *Cyfip2* from human patients or previously designed based on the crystal structure (WRC^xtal^; PDB: 3P8C) to disrupt WRC autoinhibition, are located in the tyrosine lock region and the "stem" regions (Fig. 4d, indicated by red and yellow dots, respectively)[9,27,29,30]. Single point mutation in these regions was typically sufficient to cause disease or autoactivation of WRC, suggesting the interactions in the tyrosine lock and the "stem" region are highly cooperative[9,21,28]. Releasing Y151 from the tyrosine lock would disrupt the overall conformation of the "stem" sequence, subsequently leading to WCA release and WRC activation.

The above analysis suggests Rac1 binding to the A site acts as an allosteric competitor of Y151 in the tyrosine lock. Given this model, if we disrupt the tyrosine lock, we should see enhanced Rac1 binding to the A site. To test this model, we designed three separate mutations to disrupt the tyrosine lock from different angles and then used GST-Rac1

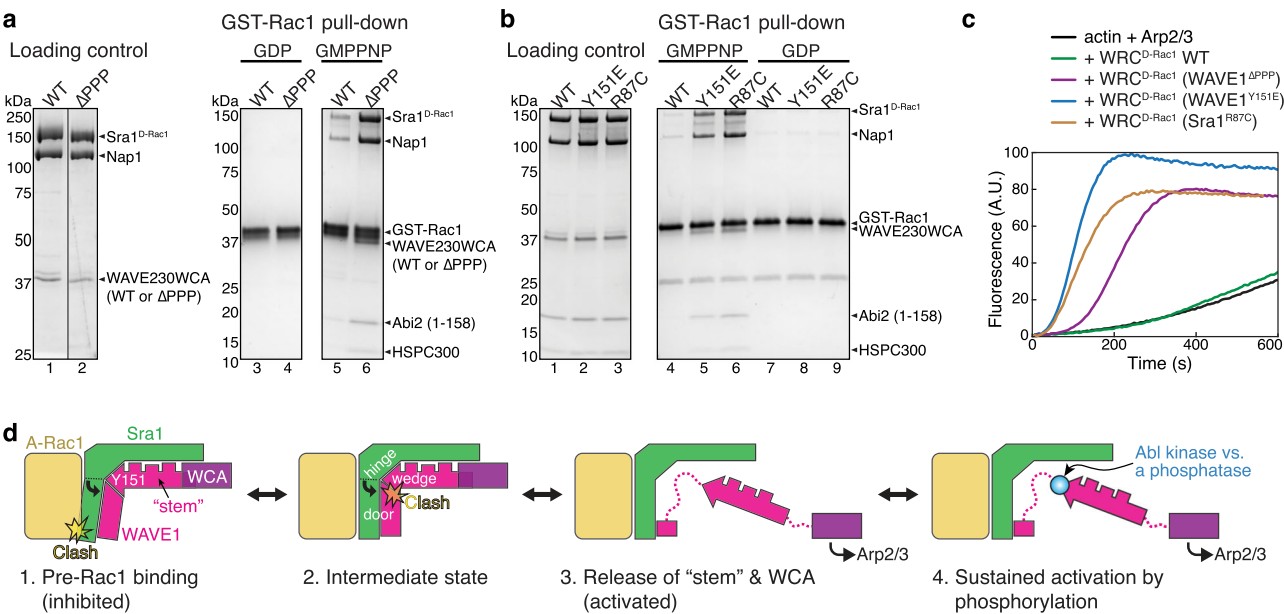

**Fig. 5 | An allosteric competition model explains WRC activation by Rac1 binding to the A site. a, b** Coomassie blue-stained SDS-PAGE gels showing GST-Rac1[P29S] (loaded with GDP or GMPPNP) pull-down of WRC[D-Rac1] bearing the indicated mutations: WAVE1[ΔPPP] in (A) (replacing [131]PPPLNI[136] with a GSGSGS linker) and WAVE1[Y151E] or Sra1[R87C] in (B). **c** Pyrene-actin polymerization assays measuring the activities of WRC[D-Rac1] used in (A-B). Reactions use the NMEH20GD buffer and contain 3.5 μM actin (5% pyrene-labeled), 10 nM Arp2/3 complex, and 100 nM WRC[D-Rac1] carrying indicated mutations. **d** A "door wedge" model describing the allosteric competition mechanism underlying WRC activation by Rac1 binding to the A site. WRC activation requires Rac1 binding to the A site to swing the door (A site in Sra1 and α4-loop-α5 in WAVE1) against the wedge (Y151 and the "stem" sequence) inserted into the door hinge. The "tug-of-war" between Rac1 binding and the tyrosine lock determines the activity level of the WRC. Phosphorylation (blue dot) of the released Y151 further shifts the equilibrium to provide an additional control of the strength and duration of WRC activation. Note that for clarity, majority of Sra1 and other WRC components not directly involved in activation are not shown in the cartoon. Source data for **a–c** are provided as a Source Data file.

to pull down various WRC[D-Rac1] that carry these mutations. Note in WRC[D-Rac1] the D site is occupied by Rac1, which allows us to specifically measure Rac1 binding to the A site. The first mutation, ΔPPP, replaces the rigid PPII helix on the WAVE1 side of the tyrosine lock with a flexible peptide linker. The second mutation is a phosphomimetic mutation, Y151E, in WAVE1. Y151 is strictly conserved from animals to plants and is known to be phosphorylated by the Abl kinase in cells to promote WRC-mediated actin polymerization and lamellipodia formation[31–33]. In a previous study, Y151E (or Y150D in WAVE2) or a mutation disrupting the binding pocket (F686E), was shown to activate the WRC both in vitro and in cells[9]. The third mutation is R87C in Sra1, which is a hotspot mutation in *Cyfip2* in human patients with DEE-65 and was shown to activate WRC in cells[27–29]. R87C should disrupt the tyrosine lock from the Sra1 side (Fig. 4c). As expected, the wild type (WT) WRC[D-Rac1] showed marginal binding to GST-Rac1 due to the low affinity of Rac1 to the A site in inhibited WRC. By contrast, all the above three mutations significantly promoted the binding (Fig. 5a, b) and caused autoactivation of WRC in pyrene-actin polymerization assays (Fig. 5c).

Together, our analysis explains how Rac1 binding to the A site promotes WRC activation through an allosteric competition mechanism. This mechanism is analogous to a "door wedge" model (Fig. 5d and Supplementary Movie 1). In the basal state, the wedge (Y151) is inserted in the door hinge (tyrosine lock) to stabilize the closed conformation. To activate WRC, Rac1 binding must push the door (the A site together with α4-loop-α5 of WAVE1) to swing around the hinge, which will pinch out the wedge, leading to the collective release of the attached "stem" region and WCA. The "tug of war" between the wedge and Rac1 binding determines the equilibrium between the closed and open conformations of the WRC and hence its activity level, while further phosphorylation of Y151 by Abl can act synergistically with Rac1 to shift the equilibrium (more details in Discussion).

## Discussion

Signaling from Rac1 GTPase to the WRC and Arp2/3 complex plays a central role in promoting actin cytoskeletal assembly in various important processes[2,5,13,19]. Nevertheless, how Rac1 binds to and activates WRC has remained a major conundrum for the last two decades. Here, our structural, biochemical, and cellular analyses have resolved this central mechanism, revealing precisely how Rac1 binds to both A and D sites through two distinct surfaces and how Rac1 binding to the A site stimulates WRC activation. Distinct from two other members of the WASP-family proteins, WASP and N-WASP, in which the Rho-GTPase Cdc42 releases the inhibited WCA through a direct competition mechanism[4], WAVE in the WRC is activated by an allosteric competition mechanism analogous to a "door wedge" model, in which Rac1 binding to the A site drives conformational changes that propagate to the tyrosine lock region to destabilize meander binding and release WCA (Fig. 5d and Supplementary Movie 1).

Among the many ligands of the WRC, Rac GTPases are the ubiquitous activator[19]. Determining the activation mechanism by Rac1 gives insights into how other WRC ligands, such as inositol phospholipids, kinases, Arf GTPases, and membrane receptors, may work together with Rac to spatiotemporally control WRC activity in diverse processes[19]. For example, phosphorylation of WAVE2[Y150] and WAVE3[Y151] (equivalent to WAVE1[Y151]) by the Abl kinase promotes WRC-mediated actin polymerization and lamellipodia formation[31–33]. Our data reveal that Abl and Rac1 utilize the same mechanism to promote WRC activation. Y151 is entirely buried in the tyrosine lock in the basal state and only becomes accessible to Abl kinases when the A site-bound Rac1 releases it. Once phosphorylated, Y151 can no longer antagonize Rac1 binding. Instead, the phosphorylated Y151 will sustain WRC activation until it is reversed by a phosphatase (Fig. 5d). The crosstalk between phosphorylation, dephosphorylation, and Rac1 binding can provide an intricate mechanism for cells to control both the strength and duration

of WRC activation. A similar mechanism was previously proposed for WASP and N-WASP[34].

The structural model also depicts how WRC can be possibly oriented on membranes to interact with other molecules. WRC has an acidic and a basic surface (Supplementary Fig. 4b). When binding to two Rac1 molecules, WRC can readily associate with acidic phospholipids on the membrane through the basic side. This orientation allows both Rac1 molecules to be anchored to the lipid bilayer through their prenylated basic tails, while the WCA can be readily released towards Arp2/3 complex in the cytoplasm. It is known that acidic phospholipids, such as PIP$_3$, cooperate with Rac1 to activate WRC[15]. The synergy could be achieved by increasing membrane recruitment of the WRC and/or simultaneously stabilizing a conformation compatible with Rac1 binding (such as capturing the released, positively charged α6 helix of the meander region).

In addition, other membrane-associated proteins are thought to cooperate with Rac1 to promote WRC activity, including many proteins that contain a short peptide motif named WIRS (WRC interacting receptor sequence)[13] and the Arf1 GTPase[12]. While the WIRS peptide does not activate WRC, sequences flanking the WIRS motif were shown to modulate WRC activity[13]. It is possible that these sequences act through secondary, weak interactions with structural elements important for Rac1-mediated activation, including the A site, the meander sequence, and the tyrosine lock. Knowing how Rac1 activates the WRC will help dissecting the contributions of other WRC ligands (e.g., Arf1 GTPase and various membrane receptors).

Our data also shed light on how various disease-related mutations influence WRC activation. Many missense mutations that cause the neurodevelopmental disorder DEE-65 are clustered around the A site and the tyrosine lock region (Fig. 4e), which was previously named the "Cyfip2 hotspot #1"[19]. Our data suggest all known mutations in this region increase WRC activity, either by destabilizing the tyrosine lock or the "stem" region (such as R87C), or by promoting Rac1 binding (such as Y108H). It is remarkable that Y108H only mildly increases the sensitivity of the WRC for Rac1, underscoring the importance of precisely controlling WRC activity in the cell. Our structure also reveals how the melanoma-causing mutation, Rac1P29S, enhances WRC activation by facilitating its binding to the A site. Together, these data emphasize the need of developing inhibitors that can target Rac1-mediated WRC activation for the treatment of related diseases.

Our data clearly support that D site Rac1 binding does not directly activate WRC. We posit that the D site Rac1 binding has at least two major functions. First, the D site can facilitate membrane recruitment of the WRC due to its high affinity. Second, D site Rac1 binding may enhance A site Rac1 binding. Several pieces of evidence support this notion. First, the previously measured binding isotherms of GST-Rac1 binding to the WRC suggested cooperativity between A and D sites[14]. Second, mutating the D site was shown to abolish WRC activation in pyrene-actin assembly assays[14]—it is possible that without D site Rac1 binding, A site affinity is too low to show activation in these assay conditions. This is consistent with our new results, showing that while the WRC with Rac1 tethered to the A site is basally active, the activity can be further promoted by D site Rac1 binding (Supplementary Fig. 3d).

Lastly, the WRCAD-Rac1 structure clearly reveals the similarities and differences between Rac1 binding to the A site of Sra1 and the recently discovered Rac1 inhibitor, CYRI-B[24,26]. CYRI-B and the A site region (designated as the DUF1394 domain) share little homology in sequence (21% identity)[25], but high similarity in structure (3.7 Å r.m.s.d.) (Supplementary Fig. 6p–s). Our structure reveals that Rac1 has a similar orientation in binding to both Sra1 A site and CYRI-B surfaces, which are both positively charged (Fig. 3b vs. Supplementary Fig. 6q). Nevertheless, except for a few conserved residues sharing a similar mechanism to bind Rac1, including the Arginine (Sra1R190 vs. CYRI-BR161)

anchoring into the negatively charged pocket on Rac1 (Fig. 3c vs. Supplementary Fig. 6s), most other interactions are largely different, with the Sra1 A site involving more extensive interactions.

In summary, our work delineates the binding and activation mechanism of WRC by Rac1 GTPase. It clearly demonstrates that Rac1's engagement to the A site and not just to the D site leads to activation of WRC and suggests possible cooperativity between the two sites. The latter needs to be clearly established by future studies. Besides providing a structural perspective for several disease mutations, this study provides a mechanistic foundation for understanding how small GTPases can trigger the activation of WRC and regulate the WRC-Arp2/3-actin signaling axis.

## Methods

### Protein purification
All WRC constructs used in this work were derived from WRCapo (also called WRC230WCA or WRC230VCA[14]) by standard molecular biology procedures and were verified by Sanger sequencing. WRC230WCA contains human full-length Sra1, full-length Nap1, WAVE1(1-230)-(GGS)$_6$-WCA(485-559), Abi2(1-158), and full-length HSPC300. Other WRCs contain modified subunits (see Supplementary Tables 1, 2 for detailed protein sequences and WRC compositions, and Supplementary Table 4 for DNA oligos that were used to create various constructs). Briefly, WRCD-Rac1 was created by tethering Rac1Q61L/P29S(1-188) to the C-terminus of Sra1 using a (GGS)$_4$ peptide linker. WRCAD-Rac1 was created from WRCD-Rac1 P29S by inserting (GGS)$_6$-Rac1Q61L/P29S(1-188)-(GS)$_6$ between Y423 and S424 of Sra1. WRCD-Rac1/ΔPPP was made from WRCD-Rac1 by replacing 137PPPLNI230 in WAVE1 with a GSGSGS linker. WRCD-Rac1/Y151E and WRCD-Rac1/R87C were made from WRCD-Rac1 by introducing Y151E and R87C to Sra1, respectively.

All WRCs were expressed and purified from the Sf9 (Thermo Fisher) and Tni (Expression Systems) insect cells (for Sra1 and Nap1) and *E. coli* (for WAVE1, Abi2, and HSPC300) through multiple chromatographic steps, essentially as previously described for WRC230WCA and WRC230ΔWCA-Rac1 [14,35]. A gel filtration step by a 24-ml Superdex200 (Cytiva) was always used as the final, polishing step of purification to exchange buffer and evaluate the purity, assembly, and potential aggregation in each preparation. All WRC constructs appeared to be properly assembled into stable complexes, behaving similarly to WRC230WCA during each step of the reconstitution, showing no noticeable signs of aggregation or misfolding (Supplementary Fig. 5). All other proteins were purified using previously established procedures, including GST-Rac1 and untagged Rac1 WT or mutants carrying P29S or P29S/Q61L, Arp2/3 complex, actin, WAVE1 WCA, TEV protease, and HRV 3 C protease[14]. Note Rac1Q61L/P29S constitutively binds to GTP without noticeable hydrolysis during long-term storage. The bound GTP cannot be exchanged with other nucleotides (such as GDP) using standard EDTA-chelating procedures even at 37 °C (Supplementary Fig. 3e). Therefore, all constructs containing Rac1Q61L/P29S were used as the GTP form without being reloaded to other nucleotides.

### Nucleotide test of Rac1 GTPases
Rac1QP and Rac1P29S (100 μl, 150-200 μM) were first loaded with GTP or GDP by using previous EDTA-chelating procedures[14]. The proteins were then desalted into the QA buffer (20 mM Tris-HCl, pH 8) through a 5-mL HiTrap Desalting column (Cytiva) and then denatured by five volumes of 8 M urea for 30 min. The denatured protein samples were filtered through a centrifugal concentrator with a molecular weight cut-off (MWCO) of 10 kDa. The collected flow-through, which contained the released nucleotides but not protein, was loaded onto a 1-mL HiTrap Q column at a flow rate of 1 mL/min and eluted with 25 mL of buffer developed over a gradient of 50–500 mM KCl. Pure GTP and GDP were injected into the column separately as controls.

## GST pull-down assay

GST pull-down assays were performed as previously described[14]. Briefly, 100-200 pmol of GST tagged proteins as bait and 100-200 pmol of WRCs as prey were mixed with 20 µL of Glutathione Sepharose beads (Cytiva) in 1 mL of binding buffer containing 10 mM HEPES pH 7, 50 mM NaCl, 5% (w/v) glycerol, 2 mM $MgCl_2$, and 5 mM β-mercaptoethanol at 4 °C for 30 min, followed by three washes, each time using 1 mL of the binding buffer. Bound proteins were eluted with GST elution buffer (100 mM Tris-HCl pH 8.5, 2 mM $MgCl_2$, and 30 mM reduced glutathione) and examined by SDS-PAGE.

## Pyrene-actin assembly assay

Actin polymerization assays were performed as previously described[14] with some modifications. Each reaction (120 µL) contained 3 − 4 µM actin (5% pyrene-labeled), 10 nM Arp2/3 complex, 100 nM WRC230WCA and its variants or WAVE1 WCA, and desired concentrations of untagged $Rac1^{Q61L/P29S}$(1-188) in NMEH20GD buffer (50 mM NaCl, 1 mM $MgCl_2$, 1 mM EGTA, 10 mM HEPES pH7.0, 20% (w/v) glycerol, and 1 mM DTT) or KMEI20GD buffer (50 mM KCl, 1 mM $MgCl_2$, 1 mM EGTA, 10 mM Imidazole pH7.0, 20% (w/v) glycerol, and 1 mM DTT). We noticed that compared to KMEI20GD buffer, NMEH20GD buffer tended to facilitate ligand-WRC interaction and actin polymerization, which allowed us to reduce protein concentration and reaction time. Pyrene-actin fluorescence was recorded every 5 s at 22 °C using a 96-well flat-bottom black plate (Greiner Bio-One™) in a Spark plate reader (Tecan), with excitation at 365 nm and emission at 407 nm (15 nm bandwidth for both wavelengths).

## Sample preparation for electron microscopy

$WRC^{apo}$ and $WRC^{D-Rac1}$ samples were diluted into a buffer containing 10 mM HEPES-KOH (pH 7.0), 50 mM KCl, 1 mM $MgCl_2$, 1 mM DTT, and 5%(v/v) glycerol to a final concentration of 1.2 µM. $WRC^{AD-Rac1}$ sample was diluted using the same buffer to 0.4 µM final concentration. 3.5 µl of each sample were applied to freshly glow discharged (using air at 19 mA for 120 s in Pelco EasiGlow (Ted Pella)) 300 mesh UltrAuFoil R1.2/1.3 holey gold grids (Quantifoils). Grids were manually blotted using Whatman 1 filter paper for ~5 s to remove excess sample and immediately plunged into liquid ethane at −179 °C. Sample vitrification were carried out using a custom-built manual plunge freezing device, and the entire process was performed in a 4 °C cold room with relative humidity maintained between 90% and 95%.

## Electron microscopy data acquisition

Cryo-EM data were acquired on a 200 kV Talos Arctica (Thermo Fisher Scientific) transmission electron microscope. Dose-fractionated movies were collected using a Falcon 3EC direct electron detector (Thermo Fisher Scientific) in electron counting mode. For all the three sample datasets, each micrograph comprised of 62 dose-fractionated movie frames acquired over 40 s. Cumulative exposure dose per micrograph are 44.06 $e^-/Å^2$ for the $WRC^{apo}$ dataset, 45.27 $e^-/Å^2$ for the $WRC^{D-Rac1}$ dataset, and 41.34 $e^-/Å^2$ for the $WRC^{AD-Rac1}$ dataset. Automated data acquisitions were performed using EPU (Thermo Fisher Scientific), and 2,913 micrographs, 2512 micrographs and 1285 micrographs were collected for the $WRC^{apo}$, $WRC^{D-Rac1}$ and $WRC^{AD-Rac1}$ datasets, respectively. All datasets were acquired at a nominal magnification of 120,000x corresponding to a physical pixel size of 0.8757 Å/pixel. Data were collected with nominal defocus varying between −0.6 and −1.2 µm for the $WRC^{apo}$ dataset, between −0.5 and −1.0 µm for the $WRC^{D-Rac1}$ dataset, and between −0.8 and −1.2 µm for the $WRC^{AD-Rac1}$ dataset. Due to preferred-orientation of Rac1 bound WRC complexes in vitreous ice, the $WRC^{D-Rac1}$ and $WRC^{AD-Rac1}$ datasets had ~15% and ~30% respectively of the micrographs collected by tilting of specimen to 36° (alpha-tilt).

## Electron microscopy data processing

For $WRC^{apo}$ dataset, beam induced motion-correction and dose-weighting to compensate for radiation damage over spatial frequencies, were perform using UCSF motioncor2 program[36] implemented in the RELION v3.0.6 image processing suite[37]. Contrast Transfer Function (CTF) parameters were estimated for the motion corrected, dose-weighted summed micrographs using Gctf[38]. 2796 micrographs with estimated defocus values below −2.1 µm, were selected for further processing. 2,006,821 particles were picked using reference-free Laplacian-of-Gaussian auto-picking program in RELION. These picked particles were extracted with a box size of 424 pixels and binned by a factor of four (to 106 pixels box, with pixel size of 3.5 Å/pixel) for 2D classification, which was performed using Cryosparc v2[39]. Multiple rounds of 2D classification were performed to remove 2D class averages containing particles which were either aggregates, or false picked features, or did not contain features with well-defined secondary structures, or were significantly off centered. Metadata for the clean stack of 957,365 particles selected from Cryosparc 2D classification were converted to the RELION star format using *csparc2-star.py* script (written by Dr. Daniel Asarnow). These particles were then re-extracted after four-fold binning from 512 pixels box (0.8757 Å/pixel) to 128 pixels box (3.5 Å/pixel) in RELION for faster computation. A D-site Rac1 tethered WRC Cryo-EM map (EMDB-6642)[14], which was rescaled and padded to 128 pixels box (3.5 Å/pixel), was used as initial 3D reference map. After one round of 3D refinement, a three-class 3D classification was performed to further sort out heterogeneity. 349,065 particles belonging to the best resolved 3D class were selected for further processing. Another round of 3D refinement was performed using these selected particles, and then the resulting particles were re-extracted unbinned after re-centering picked co-ordinates from micrographs, with a box size of 512 pixels (0.8757 Å/pixel). Multiple 3D refinements (without and with masks) were performed with these particles to improve alignments and 3D angular assignments. For accurate per-particle CTF estimation and per-particle motion correction, CTF refinement was performed, followed by Bayesian polishing within RELION v3.0.7. This resulted in improvement of resolution of the reconstructed map from 3.8 Å to 3.0 Å. A three-step CTF refinement was performed in RELION v3.1 to further improve the accuracy of CTF parameters and correct for higher-order imaging aberrations. Following this step, 3D refinement further improved the resolution of the map to 2.9 Å. To further sort out local structural heterogeneity, a three-class 3D classification without alignment (clustering) was performed, and the best resolved class with the intact complex was selected for further processing. 95,319 particles from this selected class were refined to a resolution of 3.0 Å. Local resolution for the reconstructed map was estimated using RELION v3.0.7. This indicated that the core of the $WRC^{apo}$ complex was resolved to ~2.8 Å and the most flexible regions had a resolution close to 4.5 Å. 3DFSC server[38] (https://3dfsc.salk.edu) was used for the estimation of directional Fourier Shell Correlation (FSC). Signal-subtracted focused 3D refinements were performed in RELION v3.0.7 by dividing the full complex into three slightly overlapping sub-regions. This helped in improving the map quality for the peripheral flexible regions. Focused unsharpened maps and their respective half maps were individually fitted and resampled relative to the full map in UCSF Chimera[40]. Composite half maps were generated from the half maps of the individual focused regions by using the "*vop maximum*" function within UCSF Chimera. The final sharpened composite map for the $WRC^{apo}$ complex was generated from these composite half maps by post-processing within RELION. The resulting map was sharpened with a B factor of −35 $Å^2$.

For $WRC^{D-Rac1}$, the general processing scheme is similar to the one described above for $WRC^{apo}$. 1,765,193 particles from 2,434 selected micrographs (based on CTF estimation result) were extracted with an unbinned box size of 512 pixels, and subsequently binned by a factor of four. Multiple iterations of 2D classification were performed using

Cryosparc v2. 856,797 particles from 2D classes with well-defined secondary structures were selected and were imported into RELION. These particles were re-extracted with an unbinned box of 512 pixels and further binned by a factor of four. The resulting particles were subjected to a 3-classes 3D classification. 218,612 particles from the best 3D class were selected and then re-extracted unbinned with recentering of picked coordinates, with a box size of 512 pixels. Multiple rounds of 3D refinement were performed both prior and after CTF refinement and Bayesian Polishing, leading to a reconstructed map with a resolution at 3.1 Å (at 0.143 FSC). 3D clustering (classification without particle realignment) was performed with these particles for further sorting out local heterogeneity and any mis-aligned particles. 87,810 particles were selected for subsequent 3D refinements from the best 3D class. The final reconstructed WRC$^{D-Rac1}$ map was at 3.0 Å (at 0.143 FSC) resolution. Local resolution for the reconstructed map was estimated using RELION v3.0.7. The result indicated that the core of the WRC$^{D-Rac1}$ complex was resolved to ~2.8 Å and the most flexible regions had a resolution close to 4.5 Å. 3DFSC server (https://3dfsc.salk.edu) was used for estimation of directional Fourier Shell Correlation (FSC). To improve the map quality for peripheral flexible regions of the complex, focused 3D refinement with signal-subtraction were performed as described for WRC$^{apo}$ data processing. The focused maps were then used to generate the composite map. The final map was sharpened with a B factor of −39 Å$^2$.

For WRC$^{AD-Rac1}$ dataset, complete image processing was performed using Cryosparc v2. To correct for beam-induced motion, full-frame motion correction followed by patch motion corrections were performed. CTF parameters for these micrographs were estimated using Patch CTF estimation program in Cryosparc to accurately estimate local CTF parameters. Twenty two micrographs were removed due to poor CTF fitting, and 1263 micrographs were kept for subsequent processing. Template-free Gaussian blob picker was used to pick 666,417 particles from these micrographs. Picked particles were extracted unbinned with a box size of 512 pixels, and then binned by a factor of four. One round of 2D classification was performed to remove particles from 2D classes that did not represent intact complex, aggregated, or had non-particle features. 657,065 particles from the selected 2D classes were subjected to heterogenous refinement with 3, 4, 5 and 6 classes in Cryosparc. 227,361 particles from the best 3D classes were re-extracted unbinned with a box size of 512 pixels. The resulting stack of particles were subjected to homogeneous refinement, and the resulting reconstruction reached 3.2 Å resolution. To accurately correct for local beam induced motion and per-particle CTF estimation, the selected particles were subjected to local motion correction followed by global and local CTF refinement. One round of homogeneous refinement with the optimized particle set yielded a 3D reconstruction at a resolution of 3.0 Å. A three-class ab-initio reconstruction was used to further fish-out the best particles that are representative of the full complex. 139,296 particles belonging to the best resolved class from the ab-initio reconstruction were then subjected to final 3D reconstruction (one-class heterogeneous refinement) which reached 3.0 Å resolution. Local resolution for the reconstructed map was estimated using Cryosparc v2, and showed the core of the WRCA$^{AD-Rac1}$ complex was resolved to ~2.5 Å and the most flexible region had a resolution close to 4.5 Å. 3DFSC server (https://3dfsc.salk.edu) was used for estimation of directional Fourier Shell Correlation (FSC). To improve the map quality of various flexible regions, local 3D refinements with masks applied on specific regions were performed. The resulting focused maps were then stitched as described before to generate the final map. The final map was sharpened with a B factor of −59 Å$^2$.

## Atomic model building
Crystal structure of WRC (WRC$^{xtal}$) (PDB 3P8C), and Rac1 (PDB 3SBD) were used as initial models. These models were rigid-body docked into

the corresponding regions in the reconstructed maps of WRC$^{apo}$, WRC$^{D-Rac1}$ and WRC$^{AD-Rac1}$ using UCSF Chimera. Flexible fitting of the docked models into the maps were performed using Namdinator[41] (https://namdinator.au.dk), and generated composite initial model for each complex. The missing portions in the fitted models were manually built using COOT[41]. Mutation of residues in the Rac1 molecules and bound nucleotides or nucleotide analogs and divalent cations were manually edited in COOT. These models were then subjected to repeated iterations of real-space refinement in Phenix[42] to fix geometry outliers and clash issues. Manual editing of the model in COOT further improved model accuracy. The atomic models were validated using the Molprobity server[43] (http://moprobity.biochem.duke.edu/) as well as the PDB Validation server[44] (www.wwpdb.org). Presentation of structural models and maps used Pymol[45] and UCSF ChimeraX[46].

## Cell culture and co-immunoprecipitation
B16-F1-derived *Sra1/Cyfip2* KO cells (clone #3) were previously described[21], and maintained in DMEM (4.5 g/l glucose; Invitrogen) supplemented with 10% FCS (Gibco), 2 mM glutamine (Thermo Fisher Scientific) and penicillin (50 Units/ml)/streptomycin (50 μg/ml) (Thermo Fisher Scientific). Cells were routinely transfected in 6 well plates (Sarstedt), using 1 μg DNA in total and 2 μl JetPrime per well.

pEGFP-C2-Sra1 (CYFIP1) and derived C179R and Y108H mutant constructs were described previously[21,28] and correspond to the splice variant *CYFIP1a*, sequence AJ567911, of murine origin. Various point mutations in the A site were introduced by site-directed mutagenesis. The identity of all DNA constructs was verified by sequencing.

For EGFP-immunoprecipitation experiments, B16-F1-derived cell lines ectopically expressing EGFP-tagged variants of Sra1 were lysed with lysis buffer (1% Triton X-100, 140 mM KCl, 50 mM Tris/HCl pH 7.4/ 50 mM NaF, 10 mM Na$_4$P$_2$O$_7$, 2 mM MgCl$_2$ and Complete Mini, EDTA-free protease inhibitor [Roche]). Lysates were cleared and incubated with GFP-Trap agarose beads (Chromotek) for 60 min. Subsequently, beads were washed three times with lysis buffer lacking protease inhibitor and Triton X-100, mixed with SDS-PAGE loading buffer, boiled for 5 min, and examined by Western Blotting using primary antibodies against CYFIP1/2 (Sra-1/PIR121) (rabbit polyclonal, ID: 4955-B, 1:5000 dilution)[47], Nap1 (rabbit polyclonal, ID: 4953-B, 1:5000 dilution)[47], and WAVE (rabbit polyclonal, ID: pkWAVE2, 1:1000 dilution)[21], and corresponding HRP-conjugated secondary antibodies (goat polyclonal anti-rabbit, Dianova, Cat#111-035-045, 1:5000 dilution). Chemiluminescence signals were obtained upon incubation with ECL™ Prime Western Blotting Detection Reagent (Cytiva), and recorded with ECL Chemocam imager (Intas, Goettingen, Germany).

## Fluorescence microscopy, phalloidin staining, and quantification
B16-F1-derived cell lines expressing indicated EGFP-tagged CYFIP1 constructs or untransfected control cells were seeded onto laminin-coated (25 μg/ml), 15 mm-diameter glass coverslips and allowed to adhere for about 24 h prior to fixation. Cells were fixed with pre-warmed, 4% paraformaldehyde (PFA) in PBS for 20 min, and permeabilized with 0.05% Triton-X100 in PBS for 30 s. The actin cytoskeleton was subsequently stained using ATTO-594-conjugated phalloidin (ATTO TEC GmbH, Germany). Samples were mounted using Vecta-Shield Vibrance antifade reagent and imaged using a ×63/1.4NA Plan apochromatic oil objective.

For assessment of lamellipodia formation, cells were randomly selected and categorized in a blinded manner as follows: "no lamellipodia" if no phalloidin-stained peripheral lamellipodia-like actin meshwork was visible, "immature lamellipodia" if the lamellipodia-like actin meshwork was small, narrow, or displayed multiple ruffles, and "lamellipodia" if the lamellipodia-like actin meshwork appeared to be fully developed[21].

## Statistical analysis

To assess statistical significance, one-way ANOVA with Dunnett's post-hoc test was applied to compare multiple groups with one control group. Statistical analyses were performed using Prism 6.01. An error probability below 5% ($p < 0.05$; * in Figure panels) was considered to indicate statistical significance. ** and *** indicated $p$-values ≤ 0.01 and ≤0.001, respectively.

## Reporting summary

Further information on research design is available in the Nature Research Reporting Summary linked to this article.

## Data availability

The data that support this study are available from the corresponding authors upon request. Cryo-EM reconstructed maps for WRC[apo], WRC[D-Rac1] and WRC[AD-Rac1] have been deposited in the Electron Microscopy Data Bank under accession IDs EMD-26732 (WRC[apo] structure), EMD-26733 (WRC[D-Rac1] structure), and EMD-26734 (WRC[AD-Rac1] structure), respectively. Corresponding atomic models have been deposited in the Protein Data Bank with accession IDs 7USC (WRC[apo] structure), 7USD (WRC[D-Rac1] structure), and 7USE (WRC[AD-Rac1] structure), respectively. Source data are provided with this paper.

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

## Acknowledgements

This work was supported by funding from the National Institutes of Health (R35-GM128786) and start-up funds from the Iowa State University and the Roy J. Carver Charitable Trust to B.C., by Stony Brook University start-up funds to S.C., and by the Deutsche For-schungsgemeinschaft (DFG), Research Training Group GRK2223, and individual grant RO2414/8-1 to K.R. Electron Microscopy data were collected at the Stony Brook University Cryo-EM center, which is supported by the National Institutes of Health (S10 OD012272).

## Author contributions

B.C. conceived the project and supervised cloning, mutagenesis, protein expression, purification and biochemical experiments performed by S.Y., Y.L., and A.J.B. Cryo-EM sample preparation, grid vitrification, data collection, atomic-model building were performed by B.D. under the supervision of S.C. K.R. oversaw the cell biology work and cellular imaging performed by M.S. Structural analysis were performed by B.C., B.D., S.C., and S.Y. B.C. drafted the manuscript and prepared the figures with assistance from all the authors.

## Competing interests

The authors declare no competing interests.
