## [Peer Review File · Nature Communications]

REVIEWERS' COMMENTS

Reviewer #1 (Remarks to the Author):

Ding et al. report the cryo-EM structure of the activated WAVE regulatory complex (WRC) by Rac1 binding to two distinct sites, and the comparative structures of inactive WRCs without Rac binding to one or both sites. The authors succeeded in determining the high-resolution structures by using artificial linkers designed based on the previous low-resolution structure of the inactive WRC with Rac1 bound to the D site. Despite the artificial complex, the role of specific side chains in complex formation has been carefully verified by functional biochemical and cell-based analyses. The present study gives a clear picture of the significant changes in the Sra1 conformation upon Rac1 binding to the A site and the effects on disordering the WAVE meander sequence, which triggers WRC activation. Only a few minor points should still be addressed as follows:

Lines 176-178: The consistency of the binding modes is difficult to understand in the current Extended Data Fig. 4A. This figure is confusing because it includes non-effector molecules, such as GAP and GEF.

Lines 260-261: I can not realized that "the Arginine anchor is not observed in other Rac1 binding proteins" in Extended Data Fig. 6S.

Line 520: "Y967" should be corrected to "R190".

Lines 601 and 615: references 13 and 19 seem to be identical.

Line 680-681: Should be "TEV protease and HRV 3C protease cleavage sites".

Reviewer #2 (Remarks to the Author):

High quality cryo-EM structures, mutational analysis and biochemical data show that only one of two Rac binding sites is required to activate the WAVE regulatory complex (WRC). The structures show how conformational changes coupling Rac-GTP binding to the A-site release the WAVE C-terminus, so that WCA motifs can activate Arp2/3 complex. The authors use mutations (including disease-causing mutations) to verify the importance of various residues for the biochemical interactions and the function of the WRC in live cells. The advances are substantial beyond the previous work of the senior author as a postdoc.

This work is important for understanding fundamental cellular processes including cellular and organelle movements. The overall mechanism, including the common structural basis for Abl kinase and Rac1 to activate WAVE/WRC will be interesting and informative for the wide community of cell biologists. The structural details describing Rac binding to the two sites will be interesting to GTPase experts.

The structural studies by cryo-EM depended on attaching Rac covalently to two position on the WRC: (1) fusing Rac via a linker peptide to the C-terminus of Sra1 so that it can bind stably to the A-site; and (2) inserting the Rac polypeptide into a surface loop of Sra1 so that it can bind stably to the low affinity D-site. I am surprised that the in-frame tethering to Sra1 near the D-site worked. Nevertheless, the EM structure shows Rac bound to the known D-site and biochemical experiments provide evidence that WRC with the Sra1-Rac fusion protein is functional.

The new Sra1-Rac1 fusion proteins allowed the authors to show that binding of Rac-GTP to the A-site

alone releases the WAVE C-terminus, which was shown previously to activate WAVE. Questions remain about the contribution of Rac binding to the D-site, which was shown previously to be necessary to stimulate actin polymerization.

I have a few suggestions:

Use "Arp2/3 complex" rather than "Arp2/3."

Tyrosine is not a proper noun, so "Tyrosine lock" should probably be "tyrosine lock."

A 3D model showing how Rac1 was fused to Sra1 would be helpful to readers.

The movie (370585_0_video_449685_rb4bbv) is not particularly helpful, because it zooms in on too small an area of the WRC.

Fig 5D: A summary diagram is helpful and simplified structures of the components are ok. However, the cartoon does not do justice to the findings, in part because the three components are not drawn to scale.

Response to reviewers' comments

Reviewer #1 (Remarks to the Author)

Ding et al. report the cryo-EM structure of the activated WAVE regulatory complex (WRC) by Rac1 binding to two distinct sites, and the comparative structures of inactive WRCs without Rac binding to one or both sites. The authors succeeded in determining the high-resolution structures by using artificial linkers designed based on the previous low-resolution structure of the inactive WRC with Rac1 bound to the D site. Despite the artificial complex, the role of specific side chains in complex formation has been carefully verified by functional biochemical and cell-based analyses. The present study gives a clear picture of the significant changes in the Sra1 conformation upon Rac1 binding to the A site and the effects on disordering the WAVE meander sequence, which triggers WRC activation.

Only a few minor points should still be addressed as follows:

We thank the reviewer for the supportive and insightful comments.

Lines 176-178: The consistency of the binding modes is difficult to understand in the current Extended Data Fig. 4A. This figure is confusing because it includes non-effector molecules, such as GAP and GEF.

To avoid this confusion, we have now removed the two non-effector molecules from Extended Data Fig. 4A (now named Supplementary Fig. 4a).

Lines 260-261: I can not realized that "the Arginine anchor is not observed in other Rac1 binding proteins" in Extended Data Fig. 6S.

We realized where this confusion came from. The Extended Data Fig. 6S shows the Arginine anchor is observed (instead of not being observed) in CYRI-B. To remove this confusion, we have now rearranged this sentence, now reading "*This Arginine anchor is unique to Sra1 (Fig. 3c) and CYRI-B (Supplementary Fig. 6s), as it is not observed in other reported Rac1 binding proteins (not shown) (Extended Data Fig. 6S).*"

Line 520: "Y967" should be corrected to "R190".

We thank the reviewer for pointing out this mistake. We have changed "Y967" to "R190", now reading "*F37 and R190 side chains are shown as reference points.*"

Lines 601 and 615: references 13 and 19 seem to be identical.

We thank the reviewer for capturing this mistake. We have now removed the duplicated reference 19.

Line 680-681: Should be "TEV protease and HRV 3C protease cleavage sites".

There seems to be a misunderstanding with this statement. Indeed, we intend to say, "*Tev protease, and HRV 3C protease*", instead of cleavage sites. Beside key proteins that are subject of this study, we also purified the proteases required for preparation of different proteins using methodologies described in previous work.

Reviewer #2 (Remarks to the Author)

High quality cryo-EM structures, mutational analysis and biochemical data show that only one of two Rac binding sites is required to activate the WAVE regulatory complex (WRC). The structures show how conformational changes coupling Rac-GTP binding to the A-site release the WAVE C-terminus, so that WCA motifs can activate Arp2/3 complex. The authors use mutations (including disease-causing mutations) to verify the importance of various residues for the biochemical interactions and the function of the WRC in live cells. The advances are substantial beyond the previous work of the senior author as a postdoc.

This work is important for understanding fundamental cellular processes including cellular and organelle movements. The overall mechanism, including the common structural basis for Abl kinase and Rac1 to activate WAVE/WRC will be interesting and informative for the wide community of cell biologists. The structural details describing Rac binding to the two sites will be interesting to GTPase experts.

The structural studies by cryo-EM depended on attaching Rac covalently to two position on the WRC: (1) fusing Rac via a linker peptide to the C-terminus of Sra1 so that it can bind stably to the A-site; and (2) inserting the Rac polypeptide into a surface loop of Sra1 so that it can bind stably to the low affinity D-site. I am surprised that the in-frame tethering to Sra1 near the D-site worked. Nevertheless, the EM structure shows Rac bound to the known D-site and biochemical experiments provide evidence that WRC with the Sra1-Rac fusion protein is functional.

The new Sra1-Rac1 fusion proteins allowed the authors to show that binding of Rac-GTP to the A-site alone releases the WAVE C-terminus, which was shown previously to activate WAVE. Questions remain about the contribution of Rac binding to the D-site, which was shown previously to be necessary to stimulate actin polymerization.

We thank the reviewer for the in-depth review of our work and for the useful suggestions.

I have a few suggestions:

Use “Arp2/3 complex” rather than “Arp2/3.”

We thank for reviewer for the suggestion. We have changed all “Arp2/3” to “Arp2/3 complex” in the manuscript, except for several places with limited space, such as “*Rac1-WRC-Arp2/3-actin signaling axis*” in Line 39, and several annotations inside figures.

Tyrosine is not a proper noun, so “Tyrosine lock” should probably be “tyrosine lock.”

We have taken this advice and changed “Tyrosine lock” to “tyrosine lock” in text.

A 3D model showing how Rac1 was fused to Sra1 would be helpful to readers.

We agree with the reviewer, but we have already used 3D models (in addition to 2D cartoons) in multiple places to demonstrate how Rac1 is fused or tethered to Sra1. These include the colored cryo-EM density in Fig. 1b,c, where red dots indicate the tethering points and dashed lines indicate tethering peptides. More details are provided showing 3D structures of the tethering sites in Supplementary Fig. 2b,c and Supplementary Fig. 3a,c.

The movie (370585_0_video_449685_rb4bbv) is not particularly helpful, because it zooms in on too small an area of the WRC.

We respectfully disagree with the reviewer on the usefulness of the movie. WRC activation triggered by Rac1 binding to the A site does not involve large-scale, global conformational changes of the whole complex. Instead, it involves local conformational changes propagating from the A site to the tyrosine lock and ultimately to WCA. In order to clearly visualize this local conformational propagation, it is important to focus on these particular regions of interest, instead of the entire WRC.

Fig 5D: A summary diagram is helpful and simplified structures of the components are ok. However, the cartoon does not do justice to the findings, in part because the three components are not drawn to scale.

We respectfully disagree with the reviewer on the effectiveness of the cartoon in Fig. 5d. Instead, we think the cartoon well epitomizes the main findings of how Rac1 binding to the A site activates the WRC, without distracting readers by including all other components of the WRC that are not directly involved in WRC activation. We applied the similar principle in making the conformational change movie in the previous point. For this reason, we did not include majority part of Sra1, which might have cast the impression to the reviewer that “the three components are not drawn to scale”.

In making this cartoon, we indeed attempted our best to draw all components close to scale, but we had to make some compromises to balance clarity and precision. For example, we used a relatively large blue sphere to show phospho-tyrosine, and we carved out majority of Sra1 that is not directly involved in activation.

In order to make sure readers are not confused by the scale of the three components, we have included the following sentence in the figure legend: “*Note that for clarity, majority of Sra1 and other WRC components not directly involved in activation are not shown in the cartoon.*”